# SHORTEST PATH SEARCH IN LARGE-SCALE GRAPHS WITH COX-DISTRIBUTED EDGE WEIGHTS

**Yamkin M.A.**[*]
Retail Business Modeling Department
VTB Bank (PJSC)
Moscow, Russia
Department of Software Engineering
ITMO University
Saint Petersburg, Russia

**Kugaevskikh A.V.**,[†] **Avdyushina A.E.**[‡]
Department of Software Engineering
ITMO University
Saint Petersburg, Russia

## ABSTRACT

Graphs are extensively utilized in network communications to optimize data transmission routes, in social networks to model user interactions, in data storage and processing, in artificial intelligence, and in computer graphics. Graph-based structures are particularly prominent in search algorithms, including those employed by logistics companies.

A significant challenge in the application of graph structures lies in the quest for the shortest path and the reduction of dimensionality within large-scale dynamic graphs governed by a Cox distribution, all while minimizing computational time. The Cox distribution serves as a generalization of the exponential distribution, facilitating the modeling of complex processes - such as heterogeneous inter-event times or systems characterized by varying event intensities - and yielding exclusively positive real-valued outcomes. Consequently, a vital research avenue involves the exploration of existing shortest-path algorithms in conjunction with graph dimensionality reduction techniques to identify the most efficient solutions. This paper provides a comprehensive review of the primary methodologies currently employed to address this issue.

## 1 PROBLEM STATEMENT

The selection of algorithms for the development of a high-performance system aimed at solving the shortest path problem within a large-scale random graph characterized by Cox-distributed edge weights necessitates careful consideration of several critical factors:

1. The graph in question is inherently random, thus it will be generated stochastically. Furthermore, the edge capacities (weights) will be drawn from a Cox distribution, as explicitly articulated in the problem statement. The Cox distribution serves as a generalization of the exponential distribution and is adept at modeling complex processes, such as heterogeneous inter-event times or systems exhibiting time-varying event intensities. Given that the Cox distribution characterizes the time intervals between events, all realizations are confined to positive real numbers, as negative time lacks any physical significance. Therefore, it is appropriate to employ shortest-path algorithms that are not designed to accommodate negative edge weights Yu. (1998);

2. Additionally, due to the dynamic nature of the graph, the methodologies adopted must effectively integrate its temporal dimensions during processing;

3. Achieving a high-performance algorithm mandates comprehensive and efficient data preprocessing Lu et al. (2021).

---

[*]makson.yamkin@mail.ru

[†]avkugaevskikh@itmo.ru

[‡]avdiushina@itmo.ru

The focal area of this study is the optimization of algorithms for graph data. The predominant optimization strategies encompass:

1. Reducing graph dimensionality Ma et al. (2019), Haykin (1999), Brockschmidt (2020), Saerens et al. (2004), Zhao et al. (2022), Hamilton (2020);

2. Mitigating computational load through heuristic methods and preprocessing techniques Wayahdi et al. (2021);

3. Enhancing memory utilization and data storage Sanders & Schultes (2012);

4. Implementing hardware-level optimizations to expedite computation Knickerbocker (2023);

5. Leveraging distributed computing Knickerbocker (2023).

According to various authors Knickerbocker (2023), the initial two strategies are deemed the most effective for resolving the shortest-path problem in minimal time on large-scale random graphs with Cox-distributed edge weights. Therefore, this work specifically concentrates on the following two optimization strategies for graph algorithms:

1. Mitigating computational load through heuristic methods and preprocessing techniques;

2. Reducing graph dimensionality.

The research objective is to devise an approach that achieves a harmonious balance between algorithm runtime and solution accuracy for the shortest-path problem in a randomly generated graph. The use of synthetic data was agreed upon with the project client, who specified that the edges should be generated according to the Cox distribution. Furthermore, the number of vertices and the probability of edge formation were determined in consultation with the client. Consequently, a graph comprising 10,000 vertices with an edge formation probability of 0.8 was generated.

## 2 METHODOLOGY

There are numerous approaches to optimizing the shortest path search algorithm in large-scale graphs. To identify the best method, they must be compared against a baseline algorithm that computes the exact shortest path. Dijkstra's algorithm is regarded as the benchmark, as it guarantees the exact shortest path in graphs with non-negative edge weights.

Dijkstra's algorithm is a classical method for searching the shortest path, characterized by its high time complexity. According to S. & A. (2017), its running time can be expressed by the formula (1):

$$t = O(n^2 + m), \tag{1}$$

where $n$ represents the number of vertices in the graph, $m$ denotes the number of edges, and $t$ indicates the algorithm's running time.

Note on Complexity: The time complexity formulas presented below assume standard implementations. Actual performance may vary based on specific data structures and graph density.

This algorithm systematically examines all potential paths from the source vertex to the target vertex, ultimately selecting the shortest one. A notable limitation of Dijkstra's algorithm is its inability to accommodate edges with negative weights S. & A. (2017).

It is necessary to provide a description of the methods used to optimize shortest path search in large-scale graphs with Cox-distributed edge weights. The study unfolds in two phases aimed at discerning the most effective approach for enhancing shortest-path algorithms. The study consists of two phases aimed at identifying the most effective method for improving shortest-path algorithms:

1. Analyzing techniques for migrating computational load through heuristics and preprocessing. The following methods were scrutinized:

1.1. Bellman-Ford algorithm.

The Bellman-Ford algorithm is employed to determine the shortest paths from a single source vertex to all other vertices within a weighted graph Parimala et al. (2021). As noted in Parimala et al. (2021), its time complexity is articulated by the formula (2):

$$t = O(n * m), \tag{2}$$

where $n$ signifies the number of vertices and $m$ represents the number of edges in the graph.

This algorithm is capable of processing graphs that contain edges with negative weights Parimala et al. (2021). Furthermore, it can identify the existence of negative-weight cycles; however, it is unable to compute shortest paths in graphs that include such cycles, as the cost of the path can be reduced indefinitely through repeated traversal of the cycle Parimala et al. (2021). The development of this algorithm enhances the capacity to process graphs with negative edge weights Parimala et al. (2021).

1.2. A* search algorithm.

The A* algorithm represents an enhancement over Dijkstra's algorithm, as it integrates an optimization strategy that directs the graph exploration process. As a result, A* optimizes both the cost associated with the path and the efficiency of pathfinding. It is important to note that with an admissible and consistent heuristic, A* guarantees optimality, finding the definitive shortest path. Regarding complexity, while the worst-case time complexity remains comparable to Dijkstra's algorithm ( with a binary heap), the effective runtime depends heavily on the heuristic quality. In best-case scenarios with highly accurate heuristics, the number of expanded nodes approaches linearity relative to the path length, though claiming strict universally is dependent on specific graph structures and heuristic precision. The advancement of the A* algorithm facilitates the efficient computation of the shortest path in graphs without the necessity of exhaustively traversing all vertices Wayahdi et al. (2021).

1.3. ALT algorithm El-Sherbeny (2014).

The ALT algorithm is a technique that revolves around preprocessing the graph and selecting a set of landmarks, which facilitate the construction of effective heuristics. During the preprocessing phase, a limited set of landmarks is chosen, and distances from all vertices to these landmarks, as well as from the landmarks to all vertices, are computed through both backward and forward runs of Dijkstra's algorithm. The time complexity for the preprocessing stage counts by the formula (3):

$$t = O(k * (n * log(n) + m)), \tag{3}$$

where $k$ denotes the number of landmarks, $n$ is the number of vertices, and $m$ represents the number of edges.

During the query processing phase, a modified A* algorithm is employed, utilizing a landmark-based heuristic. The worst-case time complexity for the query phase count by the formula (4):

$$t = O(n * log(n) + m) \tag{4}$$

The heuristic in ALT does not change the worst-case asymptotic complexity compared to standard A* algorithm. Yet, in practice, the number of vertices explored is significantly fewer than that in the classical Dijkstra's algorithm - particularly in graphs exhibiting geometric structures El-Sherbeny (2014). Consequently, the ALT algorithm offers substantial speed improvements while incurring relatively moderate preprocessing overhead.

1.4. Contraction Hierarchies Dibbelt et al. (2014).

Contraction Hierarchies (CH) is a method designed to expedite shortest path computation by constructing a hierarchical graph structure through the iterative "contraction" (removal) of vertices in a specific order, accompanied by the addition of "shortcuts" (shortcut edges) that retain information regarding the shortest paths traversing the removed vertices.

The preprocessing phase entails the selection of a contraction order and the execution of contraction operations for each vertex. Its time complexity is generally shown in the formula (5):

$$t = O(n * (n * log(n) + m)) \tag{5}$$

Rendering it computationally intensive - particularly for large graphs. However, this phase is conducted only once.

The query phase is executed as a bidirectional search across the hierarchy, considering solely the edges that point "upward" within the hierarchy. In the worst-case scenario, the query complexity shown in the formula (6):

$$t = O(n * log(n) + m) \tag{6}$$

Yet in practice, it approaches linearity relative to the length of the resultant shortest path, facilitating extraordinarily rapid query times. Thus, despite the high cost of preprocessing, Contraction Hierarchies achieves some of the most exceptional performance among all exact shortest path algorithms.

1.5. Bidirectional Dijkstra's algorithm Haeupler et al. (2025).

Bidirectional Dijkstra's algorithm is a variant of the classical Dijkstra's algorithm, characterized by the execution of two simultaneous searches: one proceeding forward from the source vertex and the other backward from the target vertex. The search concludes when the frontiers of the two searches converge, ensuring that the resultant path is indeed the shortest.

This algorithm does not necessitate any preprocessing of the graph. Its worst-case time complexity aligns with that of the standard Dijkstra's algorithm, expressed in the formula (7):

$$t = O(n * log(n) + m), \tag{7}$$

When utilizing a binary heap, where $n$ signifies the number of vertices and $m$ indicates the number of edges. Nevertheless, in average scenarios - particularly within sparse or geometrically structured graphs - the number of vertices visited is markedly diminished.

Based on the description of shortest path algorithms, it is necessary to assess how their theoretical foundation influences the search for the shortest path in a graph whose edges are described by a Cox distribution. The Cox distribution is characterized by its ability to model a wide range of processes due to its inherent flexibility. In the context of graph theory, this manifests itself in two key aspects: high variance in edge weights and a potentially heavy-tailed distribution Yu. (1998). High variance implies that edge weights can vary significantly, creating "problematic" edges with very large values and "beneficial" edges with small values within the graph.

Below is a comparative Table 1 illustrating how each of the algorithms discussed above handles the high variance of weights in a graph whose weights are generated according to the Cox distribution.

2. Analyzing graph dimensionality reduction techniques, including:

2.1. Random walks for constructing lower-dimensional graph representations.

The methodology is predicated on random walks across the graph. A random walk is a sequential process involving movement from one vertex of the graph to another, chosen at random Zhao et al. (2022). It commences with the selection of a random initial vertex, after which the walk progresses to one of its neighboring vertices, selected according to a specified probability. This transition probability can be uniformly assigned across all neighbors or weighted more heavily for "important" neighbors - where importance may be determined, for instance, by the edge weight Zhao et al. (2022).

This process is reiterated for a predefined number of steps. Upon completion, a subset of nodes or edges traversed during the walk is selected Zhao et al. (2022). Random walk sampling is an exceptionally efficient technique as it processes only a fraction of the graph's nodes. Furthermore, the random transition operators require minimal computational resources Zhao et al. (2022), rendering this method particularly well-suited for the problem at hand.

2.2. Principal Component Analysis (PCA) for linear transformation of the feature space.

Table 1: Comparative Analysis of Algorithms

| Shortest-path algorithm | Sensitivity to High Variance | Sensitivity to Heavy-Tailed Distribution | Justification |
|---|---|---|---|
| Dijkstra's algorithm | High | High | Early path cost estimates become inaccurate, which increases the number of vertices explored. |
| Bellman-Ford algorithm | Low | Low | Guarantees optimality even for negative weights, but the complexity is excessive for positive Cox-distributed weights, resulting in maximal runtime. |
| A* algorithm | High | High | Heuristic quality is directly dependent on the scale of the weights: may lead to a complete failure of acceleration. |
| ALT algorithm | Low | Low | A guaranteed lower bound based on the triangle inequality does not depend on the absolute values of the weights, but only on their order. |
| Contraction Hierarchies | Medium | Medium | Performance depends on the structure of the hierarchy constructed based on the order of the weights, rather than on their absolute values. |
| Bidirectional Dijkstra's algorithm | Medium | Medium | Reduces the search space through bidirectional frontier convergence; more robust to variance than standard Dijkstra due to fewer iterations. |

PCA stands as one of the foremost techniques for reducing data dimensionality. This method can also be effectively applied to graph data Saerens et al. (2004). In this study, each vertex is represented by a feature vector corresponding to its row in the weighted adjacency matrix, encoding its connectivity profile and edge weights. PCA is applied to these vectors to reduce feature dimensionality, followed by clustering algorithms to group similar vertices into super-nodes. Consequently, the 'reduced-dimensional graph' consists of these super-nodes. The shortest path is initially computed on this coarsened graph. To address the limitation of information loss, the resulting path is not directly back-projected via PCA reconstruction; instead, it serves as a guide for a refined local search within the clusters of the original graph. This ensures topological consistency while leveraging the speed of dimensionality reduction Saerens et al. (2004).

2.3. Spectral clustering.

Clustering seeks to organize similar vertices into larger "super" vertices that encapsulate the information from all vertices within the group. These methodologies facilitate the partitioning of a graph into subgraphs. "Spectral clustering utilizes the eigenvectors of the graph Laplacian matrix derived from the weight matrix to partition the graph into subgraphs. While standard spectral clustering assumes undirected graphs, we apply weight symmetrization for directed cases to ensure compatibility. Clustering organizes similar vertices into larger 'super-vertices' that encapsulate the information from all vertices within the group. Crucially, edge weights between these super-vertices are assigned as the average weight of all original edges connecting nodes between the respective clusters. This aggregation preserves the global structure while reducing computational complexity Jain et al. (1999).

## 3 RESULTS

Experiments were conducted on synthetic graphs generated according to the specified conditions. Figure 1 presents an example graph with 100 vertices, where the edge connection probability is 0.8, and edge weights are drawn from a Cox distribution. This smaller graph is illustrated for visual clarity; all algorithm evaluations were executed on a graph comprising 10000 vertices. The performance of the algorithms was assessed utilizing two key metrics: execution time (in seconds)

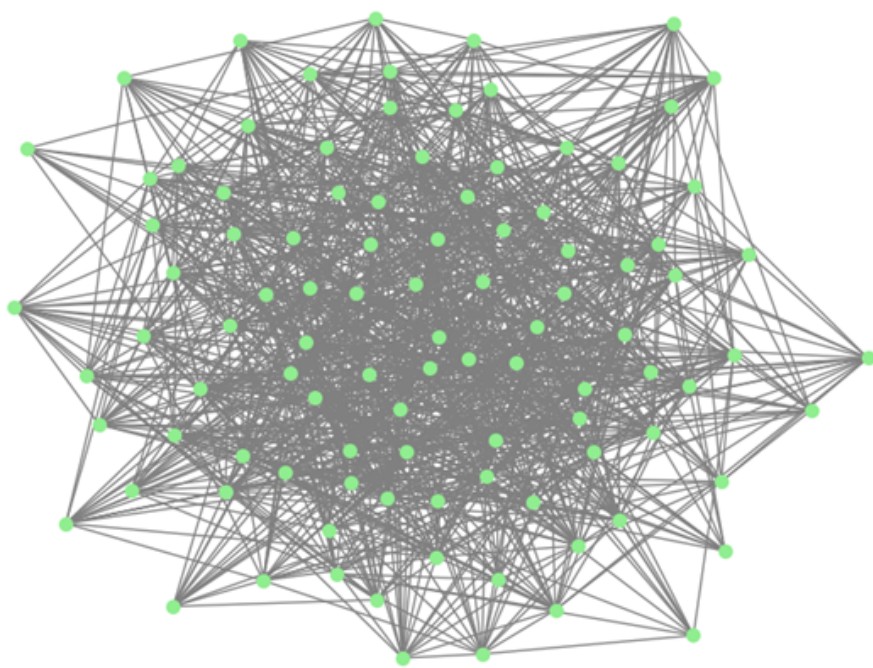

Figure 1: A synthetic graph with edge weights generated according to a Cox distribution.

Table 2: Results of shortest-path search using computational load reduction methods (mean $\pm$ std, $n = 20$)

| Shortest-path algorithm | Relative path length error, % | Path search time, s |
|---|---|---|
| Dijkstra's algorithm | $0.0 \pm 0.0$ | $25.3 \pm 1.2$ |
| Bellman-Ford algorithm | $0.0 \pm 0.0$ | $251.3 \pm 8.7$ |
| A* algorithm | $51.2 \pm 8.9$ | $0.57 \pm 0.12$ |
| ALT algorithm | $0.0 \pm 0.0$ | $0.16 \pm 0.03$ |
| Contraction Hierarchies | $0.0 \pm 0.0$ | $0.18 \pm 0.04$ |
| Bidirectional Dijkstra's algorithm | $0.0 \pm 0.0$ | $30.1 \pm 1.8$ |

Note: Results averaged over 20 runs. The random state parameter was varied from 1 to 20 to maintain the integrity of the experiment. Standard deviation reflects variability across different graph instances.

and the relative error of the computed path length compared to the solution derived from Dijkstra's algorithm.

The experiments were carried out using the following hardware: an Intel Core i5-1240P processor (12 cores, 16 threads), an integrated graphics adapter with 128 MB of video memory, and 16 GB of RAM.

The results of the experiments investigating methods for reducing computational load in shortest-path searches are summarized in Table 2.

Based on an analysis of Table 2, it can be inferred that the ALT algorithm emerges as the most optimal choice, excelling in both path search duration and the relative error of the computed path length.

Based on the fact that PCA and spectral clustering are machine learning methods highly sensitive to the initialization of initial parameters, the first stage requires selecting the hyperparameters for these methods. Specifically, the primary parameter for PCA is the number of principal components

to which the graph's dimensionality is reduced (the selection of this hyperparameter is illustrated in Figure 2). For spectral clustering, one of the key parameters is the number of clusters (the selection of this hyperparameter is illustrated in Figure 3).

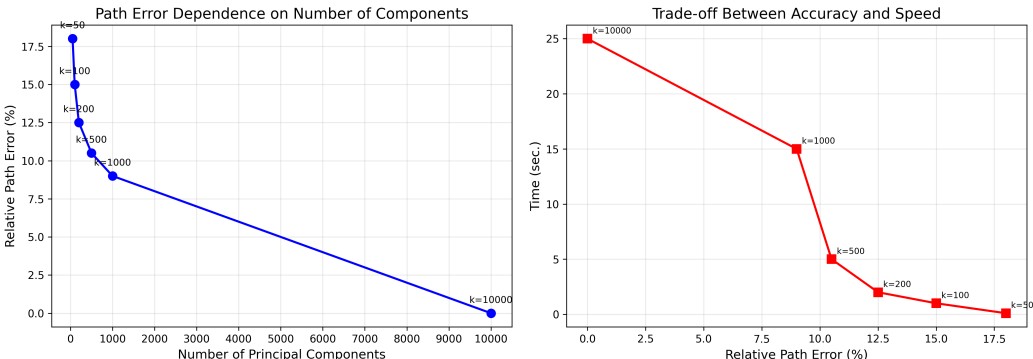

Figure 2: Determining the optimal number of principal components for graph dimensionality reduction with PCA.

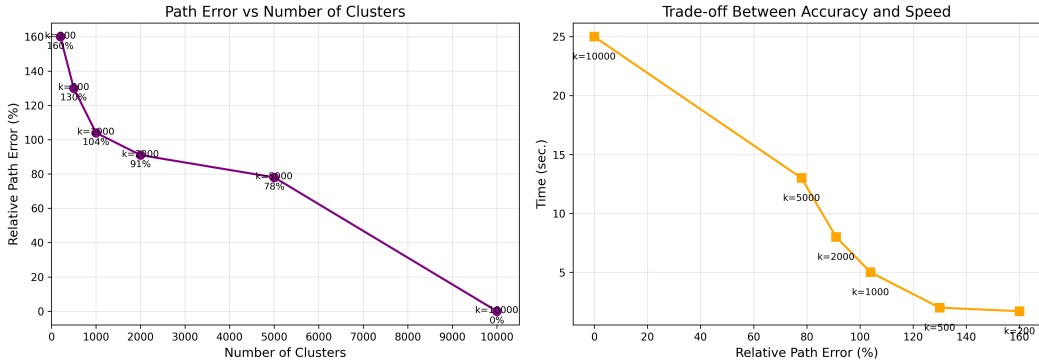

Figure 3: Determining the optimal number of clasters for graph dimensionality reduction with Spectral Clustering.

From Figure 2, it can be concluded that the optimal number of principal components is 500, as it represents a trade-off between pathfinding time and the relative error of the resulting shortest path. From Figure 3, it can be concluded that the optimal number of clusters is 1000, as it likewise represents a trade-off between pathfinding time and the relative error of the resulting shortest path. Using these optimized hyperparameter values, a series of experiments were carried out to measure both the computational time and the relative error.

Table 3 presents the results of experiments focused on graph dimensionality reduction techniques applied in shortest-path searches. Following the graph dimensionality reduction, Dijkstra's algorithm was used to find the shortest path.

An examination of Table 3 indicates that PCA stands out as the most effective method, achieving an admirable balance between path search time and relative error.

## 4  DISCUSSION

This study examined various methods for accelerating shortest path search, as well as techniques for reducing graph dimensionality. An experimental and theoretical comparative analysis of these methods was conducted. Below, the authors describe the most applicable methods for solving the

Table 3: Results of shortest-path search using graph dimensionality reduction methods (mean $\pm$ std, $n = 20$)

| Shortest-path algorithm | Relative path length error, % | Path search time, s |
|---|---|---|
| Dijkstra's algorithm | $0.0 \pm 0.0$ | $25.3 \pm 1.2$ |
| Random walks | $6.2 \pm 1.8$ | $17.1 \pm 2.3$ |
| PCA | $11.4 \pm 3.1$ | $5.2 \pm 0.8$ |
| Spectral clustering | $79.5 \pm 12.4$ | $15.3 \pm 3.1$ |

Note: Results averaged over 20 runs. The random state parameter was varied from 1 to 20 to maintain the integrity of the experiment. Standard deviation reflects variability across different graph instances.

problem of finding the shortest path in a high-dimensional graph with edge weights generated according to the Cox distribution.

In summary, among the various methods for reducing computational load, the ALT algorithm demonstrates unparalleled optimization. This algorithm is predicated on preprocessing, specifically the precomputation of landmark-based distance estimates, which facilitate a more informed and efficient search for the shortest path Wayahdi et al. (2021). Consequently, the ALT algorithm achieves minimal relative error while markedly decreasing query time in comparison to the traditional Dijkstra's algorithm. Meanwhile, the absolutely shortest path search method demonstrated the highest relative error between the found path and the true shortest path. The authors hypothesize that improving its accuracy would require the use of alternative heuristics.

Among the techniques for graph dimensionality reduction, PCA exhibits superior performance. It adeptly reduces the dimensionality of the graph, thereby enabling subsequent shortest-path computations to be executed with both high speed and minimal approximation error. Meanwhile, Spectral Clustering exhibited the highest relative error between the found path and the true shortest path. The authors suggest that clustering inevitably leads to a significant loss of information about the original graph properties, which substantially degrades the accuracy of pathfinding when this method is applied. It should be noted that the spectral clustering method, under various parameter values based on Figure 3, consistently shows low results, indicating that the method is inapplicable for solving the problem. In contrast, the PCA method, based on Figure 2, consistently shows good results with a relative error of no more than 20 percent, indicating its good applicability for solving the problem.

This study has uncovered a crucial dependence of method efficiency on the dynamic characteristics of the graph. In real-time systems characterized by frequent topology updates, adaptive mechanisms for landmark recomputation (in the context of ALT) become imperative. Promising avenues for future research include:

- It is essential to test various shortest path algorithms alongside graph dimensionality reduction methods, as their joint application may create a synergistic effect, decreasing the relative error and substantially reducing search time.

- Testing of algorithms for finding the shortest path in dynamic graphs.;

- Developing efficient algorithms for incremental landmark updates;

- Optimizing implementations through low-level programming languages and parallel computing techniques.

From the conducted work, it can be concluded that the ALT algorithm is optimal for shortest path finding in large-scale graphs with a Cox distribution, as it achieves a balance between computational efficiency and path accuracy. This method was first applied to solve the problem of finding the shortest path in a high-dimensional graph with weights generated according to the Cox distribution. In this context, the method effectively addresses the issue of high variance in edge weights arising from the Cox distribution. ALT precomputes distances from each graph vertex to a set of preselected "landmark" points. During the search, it leverages these values along with the triangle inequality to obtain a guaranteed correct lower bound on the path cost. This lower bound is always less than or equal to the true path cost; however, it does not depend on the specific weight values of the edges, but rather on their order and the structure of the graph. This makes ALT exceptionally robust to the high variance and heavy tails characteristic of the Cox distribution—a finding corroborated by

conducted experiments with the algorithm. Thus, this method can be used in practice to accelerate the shortest path search in high-dimensional graphs with edge weights generated according to the Cox distribution.

## 5 CONCLUSION

The results obtained hold substantial practical significance for the advancement of high-performance routing systems across logistics, telecommunications, and transportation infrastructure - domains where swift decision-making and precise route optimization amidst dynamically evolving network topologies are paramount. The proposed approach establishes a robust theoretical and experimental framework for constructing scalable solutions capable of processing graphs with up to 10000 vertices, achieving latency of under one second.

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
