# OpenReview forum: "Shortest Path Search in Large-Scale Graphs with Cox-Distributed Edge Weights"
_mathai.club/MathAI/2026/Conference — MathAI 2026 Conference Submission_

### Official Review · Reviewer_DH6Y · 2026-03-12
**Review of “Shortest Path Search in Large-Scale Graphs with Cox-Distributed Edge Weights”**

**Rating:** 6
**Confidence:** 3

**Review:**

This paper studies approaches to the shortest path problem in large-scale graphs where edge weights follow a Cox distribution. The authors compare several classical shortest-path algorithms and graph dimensionality reduction techniques in order to identify combinations that reduce computation time while maintaining acceptable accuracy. Experiments on synthetic graphs suggest that the ALT algorithm provides the best performance among search methods, while PCA offers a reasonable trade-off between dimensionality reduction and path accuracy.

This work investigates algorithmic strategies for computing shortest paths in large-scale random graphs with Cox-distributed edge weights. The paper focuses on two optimization directions: reducing computational load through algorithmic improvements and reducing graph dimensionality before performing path search. The study evaluates several well-known algorithms such as Dijkstra, Bellman–Ford, A*, ALT, Contraction Hierarchies, and Bidirectional Dijkstra, and also explores dimensionality reduction approaches including random walks, PCA, and spectral clustering. The experiments are conducted on synthetic graphs with up to 10,000 vertices and high edge density. According to the reported results, the ALT algorithm provides the best trade-off between computation time and accuracy, while PCA appears to be the most practical dimensionality reduction technique for the considered setup.

Overall, the paper addresses a relevant problem in graph algorithms and large-scale network analysis. The topic is interesting, especially in contexts such as routing, logistics, and communication networks where fast path computation is essential. However, the work mainly provides a comparative overview of existing methods rather than introducing a fundamentally new algorithmic contribution.

Strengths

The paper considers a practical problem related to shortest-path computation in large-scale graphs with stochastic edge weights.
The study reviews several classical algorithms and optimization strategies in a structured way.
The experimental section provides a straightforward comparison of multiple approaches using common metrics such as runtime and relative path error.
The idea of combining dimensionality reduction (e.g., PCA) with accelerated shortest-path algorithms is interesting and could be useful in practice.

Weaknesses

The paper mainly surveys and compares existing algorithms rather than proposing a new methodological contribution.
The description of the experimental setup is relatively limited (e.g., implementation details, hardware, number of runs).
The evaluation is performed only on synthetic graphs; additional experiments on real-world datasets would strengthen the results.
Some methodological explanations remain high-level and could benefit from deeper technical discussion, especially regarding how Cox-distributed weights influence algorithm behavior.
The discussion of approximation errors for dimensionality reduction methods could be analyzed more thoroughly.

In general, the paper provides a concise overview and empirical comparison of several techniques for improving shortest-path computation in large graphs with Cox-distributed weights. While the study is informative and clearly structured, its novelty is somewhat limited, as the algorithms analyzed are well-known and the experimental analysis remains relatively basic. With more detailed experiments and deeper methodological analysis, the work could become more impactful.

---

### Official Review · Reviewer_vY4X · 2026-03-13
**Relevant topic and clear empirical comparison, but limited novelty and insufficient experimental rigor**

**Rating:** 4
**Confidence:** 4

**Review:**

This paper studies shortest-path search in large synthetic graphs with positive Cox-distributed edge weights and compares several classical search algorithms and graph reduction techniques. The main reported conclusion is that ALT gives the best trade-off among the tested shortest-path methods, while PCA is the most practical reduction method. The topic is relevant, but the current version has limited novelty, an underspecified stochastic formulation, and insufficient experimental rigor.

My main concern is limited originality. The paper does not introduce a new shortest-path algorithm, a new theoretical result, or a new analysis specialized to Cox-distributed edge weights. In its present form, it is primarily an empirical comparison of standard techniques in one synthetic setting. The main conclusion, namely that ALT performs best among the tested search methods and PCA is the most practical among the tested reduction techniques, is plausible, but not especially surprising.

A second concern is that the stochastic problem is not formulated precisely enough. The paper emphasizes that edge weights are random and Cox-distributed, but it does not make fully clear what optimization problem is actually being solved. At present, the role of the Cox-distributed model remains mostly descriptive. The paper does not explain clearly why this distribution materially changes the algorithmic problem, nor how its parameters influence the relative performance of the compared methods.

A third issue is that the experimental protocol is too narrow. The experiments appear to be conducted only on synthetic graphs, essentially in a single large configuration. This is not sufficient to support broad claims. Important factors are not varied in a systematic way, including graph density, graph family or topology, Cox distribution parameters, number of landmarks in ALT, retained dimension in PCA, and sensitivity to random initialization. The paper also lacks key reproducibility details such as the number of runs, random seeds, hardware specification, implementation details, and variability measures such as standard deviations or confidence intervals. Because the graphs are synthetic and randomly generated, reporting only single runtime and error values is not enough.

Another concern is that some technical statements are inaccurate or too imprecise. Several complexity statements are presented as if they were universal, while in fact they depend on implementation details and assumptions. More importantly, the discussion of A* is problematic: with an admissible and consistent heuristic, A* remains optimal, not merely nearly optimal. Likewise, claiming a general linear runtime for A* under a proper heuristic is too strong and not justified as written. These issues reduce confidence in the methodological section and suggest that the algorithmic discussion needs closer technical revision.

There is also a noticeable gap between motivation and evaluation. The manuscript repeatedly motivates the work using dynamic graphs and time-varying behavior, yet the actual evaluation appears to be based on static synthetic graphs. If the intended application is dynamic routing, then the experimental design should explicitly evaluate graph updates over time, the cost of re-preprocessing, online query latency after updates, and robustness under changing topology or weights. Without this, the dynamic-graph motivation is not convincingly supported by the experiments.

The strengths of the paper are that it studies a practically relevant problem, the overall structure is clear and easy to follow, and it compares several classical shortest-path and reduction methods in one place. The empirical conclusion that preprocessing-based methods such as ALT are strong baselines may still be practically useful.

The weaknesses are limited novelty, an underspecified stochastic formulation, insufficient explanation of the specific role of Cox-distributed weights, narrow and insufficiently rigorous experiments, technically inaccurate or oversimplified algorithmic claims, a mismatch between the dynamic-graph motivation and the static evaluation, and the absence of experiments on real-world graph data.

To improve the paper, the authors should formulate the optimization problem more precisely, define the Cox-distributed edge-weight model explicitly, expand the experiments across multiple graph types and stochastic regimes, report aggregated statistics over repeated runs, add implementation and reproducibility details, correct the technical discussion of A* and complexity bounds, and include dynamic-graph experiments if dynamic routing is a central motivation. The contribution would also be stronger with either a new method, a stronger theoretical analysis, or a deeper empirical study.

Overall, I find the topic relevant and the manuscript reasonably clear at a high level. However, the current contribution is mostly a benchmark-style comparison of standard methods in a narrow synthetic setting, and the paper does not yet provide enough novelty, technical precision, or experimental depth to justify acceptance.

---

### Official Review · Reviewer_Ljw5 · 2026-03-13
**While the paper addresses a significant practical problem in large-scale graph analysis, it hardly fits into the scope of the conference, and methodologically the work is far from perfect.**

**Rating:** 3
**Confidence:** 3

**Review:**

This paper addresses the practical problem of accelerating shortest-path queries in large, randomly generated graphs. The authors compare two families of optimization strategies against a Dijkstra baseline: 1) computational load reduction via heuristic algorithms (A*, ALT, Bidirectional Dijkstra) and exact methods (Bellman-Ford, Contraction Hierarchies), and 2) graph dimensionality reduction via random walks, PCA, and spectral clustering. Experiments are conducted on a synthetic graph with 10,000 vertices, an edge probability of 0.8, and edge weights drawn from a Cox distribution. The results indicate that the ALT algorithm performs best among the load-reduction methods (zero error, 0.16s), while PCA is the most effective dimensionality reduction technique (9% error, 0.36s). The paper concludes by proposing a hybrid system combining PCA for dimensionality reduction followed by ALT for pathfinding.

Major remark

While the paper addresses a significant practical problem in large-scale graph analysis, it hardly fits into the scope of the conference. Although some methods from ML are used in the paper, the main goal is to solve the shortest path problem. It would be appropriate for a conference in operations research or mathematical programming, like “Mathematical Optimization Theory and Operations Research” (MOTOR)  or "International Conference on Optimization and Applications" (OPTIMA).

Minor  remarks

1.  The application of Principal Component Analysis (PCA) is not quite clear. The paper states the covariance matrix "captures the variations among graph vertices," but it does not explain what the feature vectors for the vertices are. Is it an adjacency matrix? A node embedding matrix? This is not defined. Furthermore, the claim that one can "map back" the shortest path from the PCA-reduced graph to the original graph "as PCA permits an approximate reconstruction" is unsupported.
2. The description of spectral clustering is overly simplistic. It states that it "takes into account both the directionality of vertices... and the distances between them." Standard spectral clustering uses the eigenvectors of the graph Laplacian, which is derived from the adjacency or weight matrix. It does not natively account for "directionality." The paper states Dijkstra is used "after the graph dimensionality reduction," but it is not explained how edge weights are assigned between the super-vertices.
3. The proposed "hybrid system" of PCA + ALT is not tested, so it remains a speculative idea. The mention of graph neural networks (GNNs) and RNNs in the conclusion reads as a disconnected "future work" section that is not grounded in the research presented.
4. The comparison of different methods on random data should be done using statistical significance testing (most probably, using non-parametric statistics methods).

---

### Decision · Program_Chairs · 2026-03-20

**Decision:**

Accept (Oral)

**Comment:**

On behalf of the Program Committee of the International Conference on Mathematics of Artificial Intelligence (MathAI 2026), we are pleased to inform you that your paper has been accepted for an oral presentation at MathAI 2026.

Your paper was evaluated through a rigorous two-stage review process involving both automated screening and expert review by members of the Program Committee. The reviewers recognized the quality and contribution of your work.

Presentation details:

Format: Oral presentation (15–20 minutes + 5 minutes Q&A)
Mode: You may present either in person (offline) at the conference venue in Sirius, Russia, or remotely via Zoom. Please indicate your preferred mode when confirming your participation.
Conference dates: Marh 30 - April 3, 2026
Website: https://mathai.club
Next steps:

Please confirm your participation and presentation mode by replying to this email mathai.club@yandex.ru no later than March 15, 2026 18:00 Moscow time.
If you plan to attend in person, the organizing committee will provide accommodation details separately.
Please prepare your final camera-ready manuscript according to the formatting guidelines available at https://mathai.club and upload it to OpenReview by March 15, 2026 18:00 Moscow time.
Should you have any questions regarding the program, logistics, or your presentation slot, please do not hesitate to contact us.

We look forward to your contribution to MathAI 2026.

With kind regards,

MathAI 2026 Program Committee International Conference on Mathematics of Artificial Intelligence https://mathai.club OpenReview: https://openreview.net/group?id=mathai.club/MathAI/2026/Conference Telegram: https://t.me/MathAI_club Email: mathai.club@yandex.ru